# Enhancing titres of therapeutic viral vectors using the transgene repression in vector production (TRiP) system

H.E. Maunder[1], J. Wright[1], B.R. Kolli[1], C.R. Vieira[1], T.T. Mkandawire[1,†], S. Tatoris[1,†], V. Kennedy[1], S. Iqball[1], G. Devarajan[1], S. Ellis[1], Y. Lad[1], N.G. Clarkson[1], K.A. Mitrophanous[1] & D.C. Farley[1]

A key challenge in the field of therapeutic viral vector/vaccine manufacturing is maximizing production. For most vector platforms, the 'benchmark' vector titres are achieved with inert reporter genes. However, expression of therapeutic transgenes can often adversely affect vector titres due to biological effects on cell metabolism and/or on the vector virion itself. Here, we exemplify the novel 'Transgene Repression In vector Production' (TRiP) system for the production of both RNA- and DNA-based viral vectors. The TRiP system utilizes a translational block of one or more transgenes by employing the bacterial tryptophan RNA-binding attenuation protein (TRAP), which binds its target RNA sequence close to the transgene initiation codon. We report enhancement of titres of lentiviral vectors expressing Cyclo-oxygenase-2 by 600-fold, and adenoviral vectors expressing the pro-apoptotic gene Bax by >150,000-fold. The TRiP system is transgene-independent and will be a particularly useful platform in the clinical development of viral vectors expressing problematic transgenes.

[1] Research Department, Oxford BioMedica Ltd., Windrush Court, Transport Way, Oxford OX4 6LT, UK. † Present addresses: Wellcome Trust Sanger Institute, Wellcome Genome Campus, Hinxton, Cambridge CB10 1SA, UK (T.T.M.); AstraZeneca, Innovative Medicines and Early Development, Personalised Healthcare and Biomarkers, KC721/02, Pepparedsleden 1, 431 83 Mölndal, Sweden (S.T.). Correspondence and requests for materials should be addressed to D.C.F. (email: d.farley@oxfordbiomedica.co.uk).

The use of engineered viruses to deliver transgenes for therapeutic effect is wide-ranging. Contemporary gene therapy vectors based on RNA viruses such as γ-Retroviruses and Lentiviruses[1,2], and DNA viruses such as Adenovirus[3] and Adeno-associated virus (AAV)[4] have shown promise in a growing number of human disease indications. These include *ex vivo* modification of patient cells for hematological conditions[5,6], and *in vivo* treatment of ophthalmic[7], cardiovascular[8], neurodegenerative diseases[9] and tumour therapy[10]. Immunogen-encoding vectors based on Poxviruses[11] and Avian viruses[12] are widely used in human and animal vaccinations.

A major goal for the field of viral vector manufacturing is the improvement of production cell output during upstream processing. One of the limiting factors of attaining high-starting (crude) vector titres can be the constitutive and often high-expression of the transgene protein within production cells. For many viral vector systems, the 'benchmark' crude titres are those achieved with vectors encoding 'inert' reporter genes, such as green fluorescent protein (GFP). However, the biological activity of a therapeutic protein may impact on the production cells' metabolism/viability and/or the assembly/activity of vector virions, leading to reduced crude titres that may be several orders of magnitude lower than the benchmark. Transgene protein(s) can also be incorporated or associated with the vector, which may affect downstream purification and potentially be presented to the immune system upon vector administration. This latter point may have more bearing in pre-clinical studies during delivery of human genes to immune-competent animals, although exposure of human subjects to therapeutic proteins containing artificial linkers/domains also may not be ideal.

Typically, powerful promoters are used to drive the transgene transcription unit; these may be cellular promoters such as the human phosphoglycerate kinase promoter (huPGK) or heterologous viral promoters such as from Cytomegalovirus (CMV), or be inherent to the virus platform[13]. While the use of tissue-specific promoters (that are less active in production cells) can be employed to address this problem[14], they are often not available, or are not used due to low or unpredictable activity, when translating from animal to human studies during vector development.

We have developed the TRiP system for the universal production of DNA- and RNA-based viral vectors that encode transgenes whose activity is detrimental to vector titres. The TRiP system exploits the *Bacillus subtilis* protein Tryptophan RNA-binding Attenuation Protein (TRAP) and a TRAP-binding sequence (tbs) inserted upstream of the transgene open-reading-frame (ORF), which acts to repress translation. The natural function of TRAP is to control the tryptophan synthase pathway in bacteria[15]. This is achieved through both transcriptional- and translational-attenuation mechanisms upon stimulation of TRAP binding to the tbs in the excess of L-tryptophan (see Fig. 1 for an overview). Most TRAP homologues form a stable 11-mer ring, around which the tbs is bound. The tbs has a general consensus of $[KAGN_{2-3}]_{9-11}$ *in vivo,* although the number of 'N' spacer nucleotides can be intermittently greater than this[16]. The mechanism of translational-modulation involves the binding of TRAP to the tbs, which causes rearrangement in the RNA secondary structure in the adjacent sequence of the leader, resulting in occlusion of the Shine–Dalgarno sequence and inhibition of translation initiation. A substantial body of research exists on the characterization of natural TRAP function, with a particular focus on use of *in vitro* techniques[16–19]. This has revealed an extremely high-affinity of TRAP for an optimal tbs $[KAGNN]_{11}$ in the nanomolar range. Consequently, this led others to apply this stable TRAP/tbs complex as a positive control in development of a screening tool for novel RNA-binding proteins in mammalian cells[20].

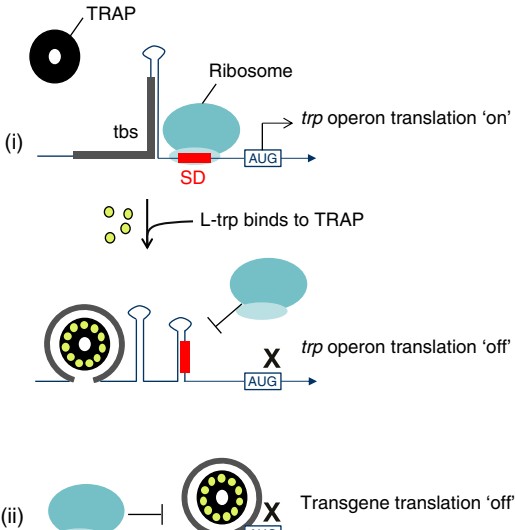

**Figure 1 | Overiew of translational block of the *trp* operon in *B. subtilis* by tryptophan RNA-binding attenuation protein (TRAP).** (i) TRAP is an 8 kDa protein that associates into a homomeric, torroidal ring comprising 11 TRAP monomers. The *trp* operon RNA leader contains a TRAP binding sequence (tbs) of KAG repeats separated by two/three, or infrequently more than three spacing nucleotides that contribute to secondary structure within the leader; this results in exposure of the Shine–Dalgarno (SD) sequence and consequently efficient translation of the *trp* operon mRNA. When L-tryptophan is present in excess, each of the 11 TRAP monomers binds L-tryptophan resulting in a conformational change on the outside of the 11-mer ring such that it can bind tightly to the tbs. This re-orders the secondary structure within the leader, causing the SD to become unavailable leading to attenuation of translation initiation. (ii) The TRAP-tbs configuration used by Nie and Htun to exemplify a RNA-binding protein-identification screen[20], and in this study for blocking of transgene translation in viral vector production cells.

Here, we fully develop a mammalian cell expression system for TRAP-mediated repression of 'problematic' transgene expression in viral vector production cells. We further demonstrate that the TRAP/tbs complex can be used to repress cap-independent translation, allowing multicistronic ORFs to be simultaneously repressed in a single cassette. Depending on the transgene, this allows substantial improvements in vector output titres, in some cases close to or at the benchmark, as demonstrated by proof-of-principle examples for the 'TRiPLenti', 'TRiPAdeno' and 'TRiPAAV' systems. Finally, deeper comparative analysis of protein profiles of lentiviral vector preparations made under standard versus TRiP approaches by mass spectrometry spectral index-normalized quantitation (MS-SINQ) suggests that downstream processing and/or quality of vector drug product may also benefit from repression of transgene(s).

## Results

**TRAP blocks single- or multi-cistronic mRNA translation.** To create the mammalian cell-based TRiP system, we codon-optimized the sequence of TRAP from *Bacillus subtilis* for human codon bias and generated a simple gene-expression cassette (Fig. 2a). To report potential transgene repression by TRAP, a tbs comprising the consensus $[KAGNN]_{11}$ was cloned within the 5′UTR of a GFP-reporter construct (Fig. 2a). Co-transfection of HEK293T cells with either TRAP plasmid or stuffer DNA control, together with pCMV-GFP or pCMV-tbsGFP was performed, and cells analysed by flow cytometry two days later (Fig. 2b,c).

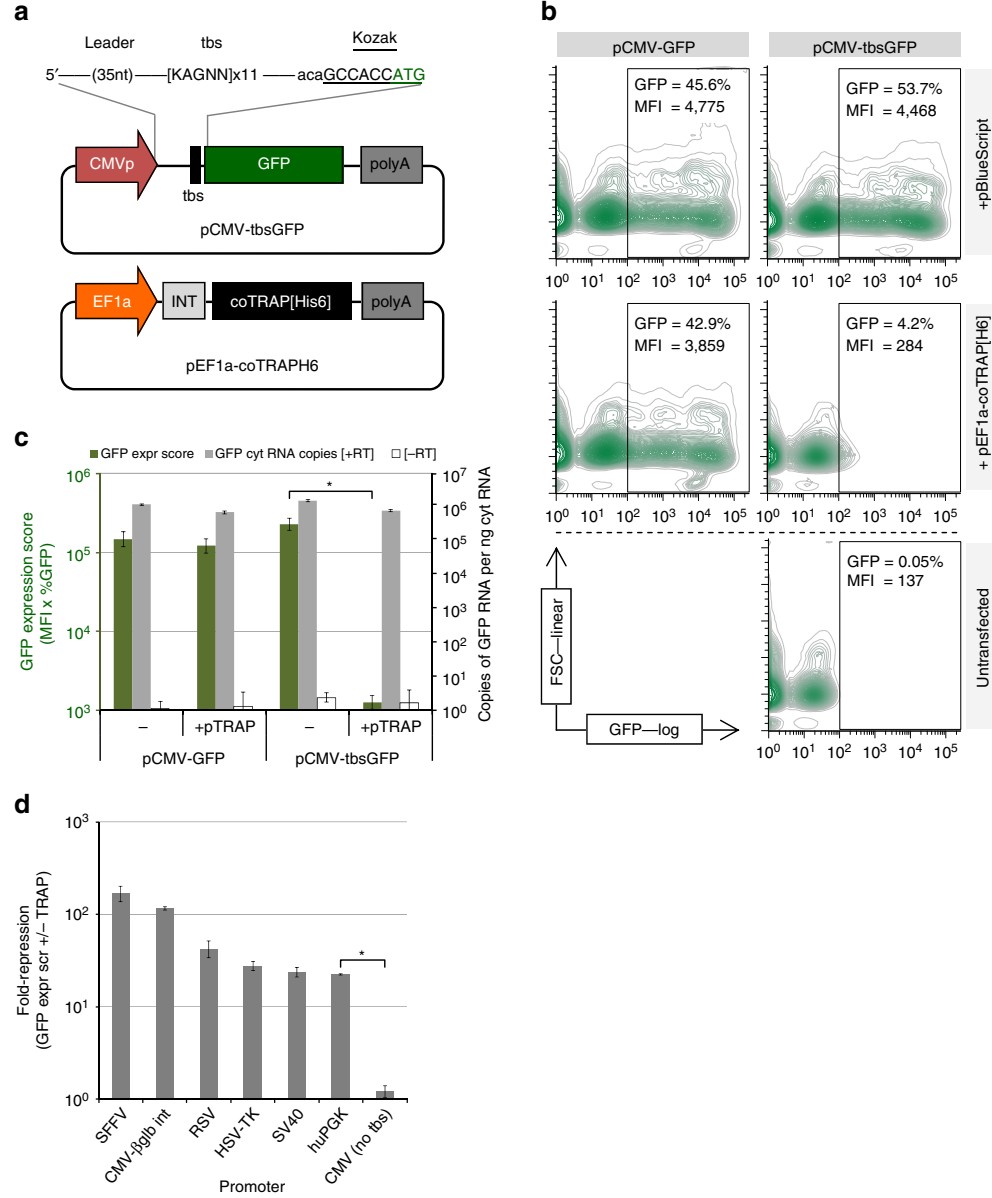

**Figure 2 | The TRAP-tbs configuration confers a translational block resulting in potent repression of GFP expression in HEK293T cells.** (**a**) The GFP reporter plasmid containing the tbs within the 5′UTR of the transcription unit and the TRAP-expression plasmid used throughout the study. *B. subtilis* TRAP was codon-optimised for human cell expression and C-terminally Hisx6 tagged. (**b**) 2-D plots (FSC v FL1/GFP) of HEK293T cells co-transfected with GFP reporter plasmids and TRAP or control plasmids, analysed by flow cytometry (MFI, median fluoresence intensity [arbitrary units]). (**c**) GFP Expression scores for the co-transfections were generated (percent GFP × MFI; plotted on left y-axis) and qRT-PCR data of cytoplasmic GFP RNA copies detected plotted on the right y-axis; grey bars—RT-positive, white bars—no RT (residual pDNA control). All data are mean average values ± s.d. [$\log_{10}$-transformed data] ($n = 8$); *$P < 1.7 \times 10^{-17}$ [Welch's *t*-test]. (**d**) Fold repression of GFP Expression Scores (difference in ± TRAP) of other promoters tested in the 'promoter-tbs-GFP' configuration in co-transfected HEK293T cells ± pEF1a-coTRAP[H6] (note that total fold repression is related to promoter strength due to a greater differential between 'on' levels compared to untransfected cells). Data are mean average values ± s.d. [$\log_{10}$-transformed data] ($n = 3$); *$P < 0.001$ [Welch's *t*-test]. (CMVp, cytomegalovirus promoter; EF1a, Elongation factor 1 alpha promoter; INT, synthetic intron; tbs, TRAP-binding sequence; coTRAP[Hisx6], codon-optimized tryptophan RNA-binding attenuation protein with C-terminal Histine tag; polyA, polyadenylation signal; SFFV, spleen focus-forming virus promoter; CMV + betaglb intron; CMV promoter with human beta globin intron; RSV, Rous Sarcoma Virus promoter; HSV-TK, Herpes simplex virus thymidine kinase promoter; SV40, Simian virus 40 promoter; huPGK, human phosphoglycerate kinase 1 promoter). are representative of two independent experiments.

In addition, total cytoplasmic RNA was extracted in parallel and qRT-PCR performed to quantify GFP sequences (Fig. 2c). Similar numbers of GFP-positive cells and fluorescence intensities were achieved in transfections without TRAP; however, these were greatly reduced specifically for only pCMV-tbsGFP in the presence of TRAP (Fig. 2b). To quantify the effect of TRAP on GFP expression within transfected cultures, we generated a GFP 'Expression Score' (median fluorescence intensity [MFI] × percentage GFP-positive cells) in a similar manner to other reports[21], which was consistent with GFP immunoblot data (Supplementary Fig. 1). The Expression Score for the pCMV-tbsGFP plus TRAP combination was 100-fold lower than

for the other conditions (Fig. 2c). In contrast, GFP messenger RNA (mRNA) levels were similar across all conditions (Fig. 2c), which confirmed that the TRAP-tbs complex was mediating a translational block, rather than affecting transgene mRNA levels. TRAP-tbs mediated repression was also achieved with other commonly used promoters including huPGK (Fig. 2d).

We investigated whether the TRAP-tbs paradigm could be applied to translation repression of multiple open-reading-frames (ORFs) encoded within transgene cassettes. We constructed bicistronic plasmids wherein the transgene transcription unit was driven by the CMV promoter and encoded luciferase at the first position (ORF1) and GFP downstream of an internal ribosomal entry site (IRES) element (ORF2). These constructs differed in that they were fully, individually or not controlled by TRAP-tbs at the ORF1 and ORF2 positions (Fig. 3a), enabling both cap-dependent and cap–independent mechanisms of translation initiation to be tested. In HEK293T cells, co-transfection experiments demonstrated that expression of multiple ORFs can be simultaneously repressed by TRAP, only when the tbs is placed upstream of the translation initiation site; this occurred for both cap-dependent and cap–independent mechanisms of expression (Fig. 3b). Both luciferase and GFP expression were repressed by TRAP-tbs by over two orders of magnitude. The capability of the TRiP system to repress IRES-driven transgene expression will be of particular benefit to retrovirus-based vector production systems because IRES-dependent transgene expression from full length human immunodeficiency virus 1 (HIV-1)-based vector genome RNA (vRNA) was found to constitute 20–30% of total transgene protein (Supplementary Fig. 2a,b). This is conceptualized in Fig. 4a, which describes the components of the TRiPRetro/TRiPLenti systems.

### Enhanced lentiviral vector titres using the TRiPLenti system.
Common methods of viral vector manufacture include the transfection of mammalian cell lines with vector DNA components, followed by a limited incubation period and then harvest of crude vector from culture media and/or cells[22]. To evaluate the impact of transgene expression on vector production, a simple therapeutic/reporter genome mixing experiment can reveal the extent of the likely impact of therapeutic protein expression on vector production (Supplementary Fig. 3). Our experience with lentiviral vector development indicates that a high-proportion of therapeutic proteins impact on crude vector titres to a measurable level, but that the contribution of transgene protein activity to loss in vector titre may be full or only partial, that is, other factors such as genome size/sequence may also contribute.

During the early stages of development of the TRiPLenti system, we sought to determine any potential adverse impact of the TRAP-tbs interaction on vector activity (for example, blocking of reverse transcription) since the full length vRNA also contains the tbs. We produced HIV-CMV-GFP and HIV-CMV-tbsGFP vectors using either the TRiPLenti system or the standard approach, and generated GFP Expression Scores in production cells, as well as vector titres (Fig. 4b). Approximately 100-fold repression of GFP expression was observed in production cells expressing pHIV-CMV-tbsGFP plus TRAP, but surprisingly the vector titres generated were not significantly different under both approaches. Since GFP is well known not to impact on vector production, this indicated that the TRAP-tbs configuration did not adversely affect vector biomechanics *per se*. It also demonstrated that transgene expression in target cells was not affected by TRAP/tbs, as the biological measure of titre (vector titre, Fig. 4b) was similar to the integrating units as determined by qPCR (Fig. 4c).

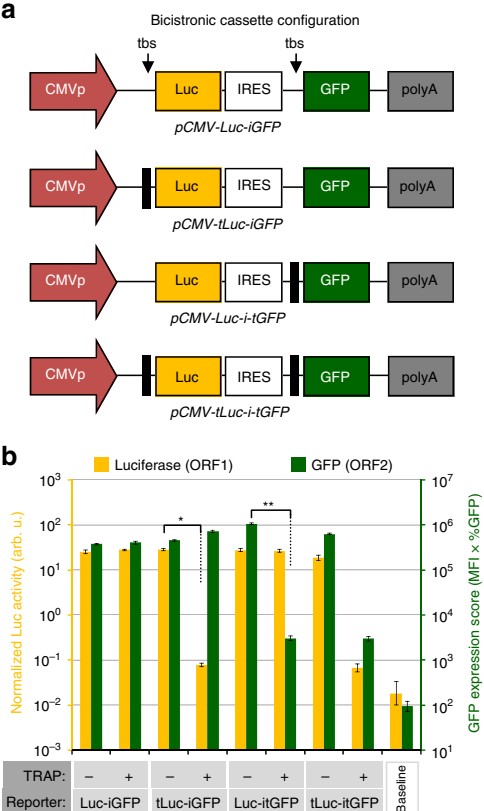

**Figure 3 | Repression of multicistronic transgene cassettes using the TRAP-tbs configuration.** (**a**) Bicistronic reporter plasmids encoding Firefly luciferase (Luc; ORF1 position) and green fluorescent protein (GFP; ORF2 position); ORF2 being dependent on cap-independent translation. The constructs vary in position and number of tbs sequences controlling the ORFs. (**b**) HEK293T cells were transfected with the bicistronic plasmids ± TRAP, and a Renilla Luciferase plasmid was co-transfected in all conditions to allow normalization of FireFly luciferase activity. Duplicate sets of cells were either analysed for normalized luciferase activity in cell lysates (left y-axis; yellow bars) or live cells analysed for GFP expression by flow cytometry (right axis; green bars). All data are mean average values ± s.d. [log₁₀-transformed data] ($n = 3$); $*P < 1.5 \times 10^{-7}$, $**P < 1.4 \times 10^{-5}$ [Welch's *t*-test]. (CMVp, Cytomegalovirus promoter; tbs, TRAP-binding sequence; IRES, internal ribosomal entry site (encephalomyocarditis virus); polyA, polyadenylation signal). Results are representative of two independent experiments.

To demonstrate the broad utility and potency of the TRiPLenti system, we produced HIV-1 and equine infectious anemia virus (EIAV)-based lentiviral vectors expressing therapeutic transgenes, some of which were found to be extremely detrimental to production titres. First, we generated the vector HIV-CMV-tbsCAR5T4 (an early stage vector encoding a chimeric antigen receptor directed to the 5T4 tumour antigen[23]) in HEK293T cells with or without TRAP, and titrated them by vector-DNA integration assay. Under the standard production method, HIV-CMV-tbsCAR5T4 titre was 300-fold lower than the HIV-CMV-GFP benchmark. However, in the presence of TRAP the titre was rescued by 30-fold (Fig. 4c).

We then used a vector under development for the treatment of glaucoma, which expresses Cyclo-oxygenase-2 (COX-2) to reduce intra-ocular pressure[24]. EIAV-CMV-COX2 and EIAV-CMV-tbsCOX2 vectors were produced with or without TRAP in HEK293T cells or in HEK293T cells stably expressing

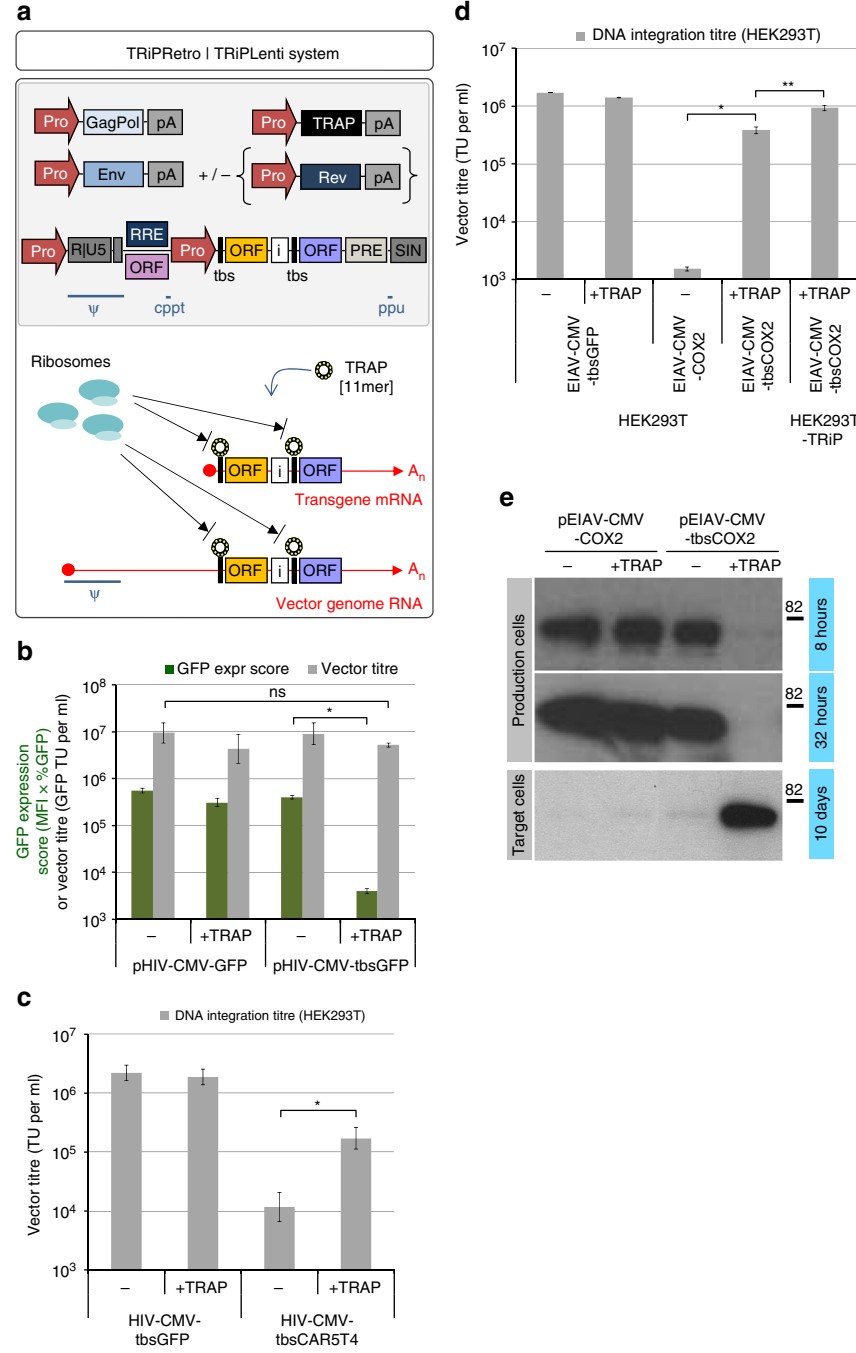

**Figure 4 | Enhanced crude titres of therapeutic lentiviral vectors using the TRiPLenti system. (a)** The TRiPRetro/TRiPLenti system requires: a tbs-modified vector genome, gag/pol, envelope and TRAP expression cassettes. HIV-1-based vector systems additionally require RRE/rev for export of genomic vector RNA (vRNA), whereas Equine Infectious Anemia Virus [EIAV] vRNA can be made rev-independent by use of an upstream open-reading frame (ORF) (unpublished). The TRAP-tbs configuration is capable of repressing cap-dependent and –independent transgene expression from full length vector genome RNA and the 'internal' transcript. **(b)** GFP Expression Scores in production cells (green bars) and crude vector titres on HEK293T cells (grey bars) of GFP-expressing HIV-1-based vectors. Data are mean average values ± s.d. [$\log_{10}$-transformed data] ($n = 3$); *$P < 3.2 \times 10^{-6}$, ns$P < 0.17$ [Welch's $t$-test]. **(c)** DNA-integrating titres of HIV-CMV-tbsCAR5T4 and HIV-CMV-tbsGFP vectors produced ± TRAP in HEK293T cells. Data are mean average values ± s.d. [$\log_{10}$-transformed data] ($n = 4$); *$P < 4.2 \times 10^{-4}$ [Welch's $t$-test]. **(d)** DNA-integrating titres of EIAV-based vectors produced ± TRAP in HEK293T cells; EIAV-CMV-COX2 and EIAV-CMV-tbsCOX2 vectors were made without or with TRAP respectively, representing standard versus 'transient' TRiPLenti protocols. EIAV-CMV-tbsCOX2 vector was additionally made in cells stably expressing TRAP, representing the 'stable' TRiPLenti protocol. Data are mean average values ± s.d. [$\log_{10}$-transformed data] ($n = 3$); *$P < 1.3 \times 10^{-5}$, **$P < 3.2 \times 10^{-4}$ [Welch's $t$-test]. **(e)** HEK293T cell lysates from EIAV-CMV-[± tbs]COX2 vector production, as well as end-of-integration assay cell lysates, were analysed by immunoblotting to COX-2 (molecular weight marker in kilodaltons). (Pro, promoter; Ext Pro, external promoter; Int Pro, internal promoter; polyA, polyadenylation signal; Ψ, packaging signal; RRE, rev-responsive element; cppt, central polypurine tract; tbs, TRAP-binding sequence; IRES, internal ribosomal entry site (encephalomyocarditis virus); PRE, post-transcriptional regulatory element; ppu, polypurine tract; SIN, self-inactivating; An, poly-adenines). Results are representative of two independent experiments.

TRAP, and COX-2 levels in replicate cultures were analysed at early and late time points during production (HEK293T cells only). Equal volumes of crude vector were titrated by DNA integration assay (Fig. 4d) and end-of-assay cell lysates analysed for COX-2 protein content (Fig. 4e). The titre of EIAV-CMV-COX2 vector made under the standard production method was three orders of magnitude lower than the EIAV-CMV-GFP benchmark vector. In contrast, EIAV-CMV-tbsCOX2 vector titres were rescued by 100-fold and 600-fold in the 'transient' and 'stable' versions of the TRiP system, respectively. COX-2 protein levels were highly repressed throughout production, but in target cells were proportional to the vector titres (Fig. 4e).

We also performed a vector mixing experiment with a second problematic therapeutic EIAV vector expressing Factor-VIII (known to inhibit VSV-G envelope incorporation into EIAV virions[14]), which confirmed that the stable version of the TRiP system allowed further improved recoveries of vector titres (Supplementary Fig. 4a–c). These data indicate that in the stable TRiP system, the pre-existing pool of TRAP within the production cell maximizes the opportunity for transgene mRNA to be repressed, leading to further benefits in titre recovery. Further examples of TRiPLenti vectors are displayed in Supplementary Fig. 4d.

**Improved protein profiles of TRiPLenti vectors encoding COX2.** Expression of the COX-2 protein during EIAV vector production drastically impaired vector titre. To assess the impact on viral particle protein content, we generated concentrated, serum-free preparations of EIAV-CMV-GFP, EIAV-CMV-COX2 and EIAV-CMV-tbsCOX2, all produced in the presence of TRAP. Analysis by MS-SINQ allowed the relative quantification of viral and cellular proteins within preparations, and hits that varied by >4-fold between replicates were excluded from subsequent data analysis. Supplementary Fig. 5a displays average data for the top 154 proteins in EIAV-CMV-GFP vector preparations, representing ~90% of protein abundance. Gag and Pol were detected close to the expected 20:1 ratio, as well as cellular proteins documented to be incorporated into HIV-1 virions[25,26]. The key differences between the three vector preparations showing both common and uncommon proteins are portrayed graphically in Fig. 5, indicating that EIAV-CMV-GFP and EIAV-CMV-tbsCOX2 had similar profiles in contrast to EIAV-CMV-COX2, which had a large number of low abundant proteins making up its profile 'tail'. We performed a three-way comparison of common proteins (plus COX-2) detected in all three vector types, normalizing abundance to that of the EIAV-CMV-tbsCOX2 vector. This allowed comparison of the effects of expression of COX-2 or GFP, modelling 'active' or 'inert' proteins, respectively, and are summarized in Table 1, which presents select data from both the first (duplicate) and second (quadruplicate) independent experiments. COX-2 content in EIAV-CMV-COX2 was more abundant than Gag protein, and was almost 3,000-fold greater in abundance than COX-2 levels in EIAV-CMV-tbsCOX2 in the first experiment and COX2 was not detectable in EIAV-CMV-tbsCOX2 vectors in the second experiment. A number of observations were made that suggest that the 2–3 log difference in transducing activity between the two COX-2 vectors (Fig. 4d) is multifactorial. COX2 expression mediated both a minor quantitative effect on virion abundance (Gag; ~5-fold) and a major qualitative effect on virion activity, namely reduced VSV-G incorporation (~50-fold). These data were consistent with the other physical analysis of vector preparations performed, namely F-PERT (measures vector virion RT activity) and immunoblot analysis for capsid and VSV-G

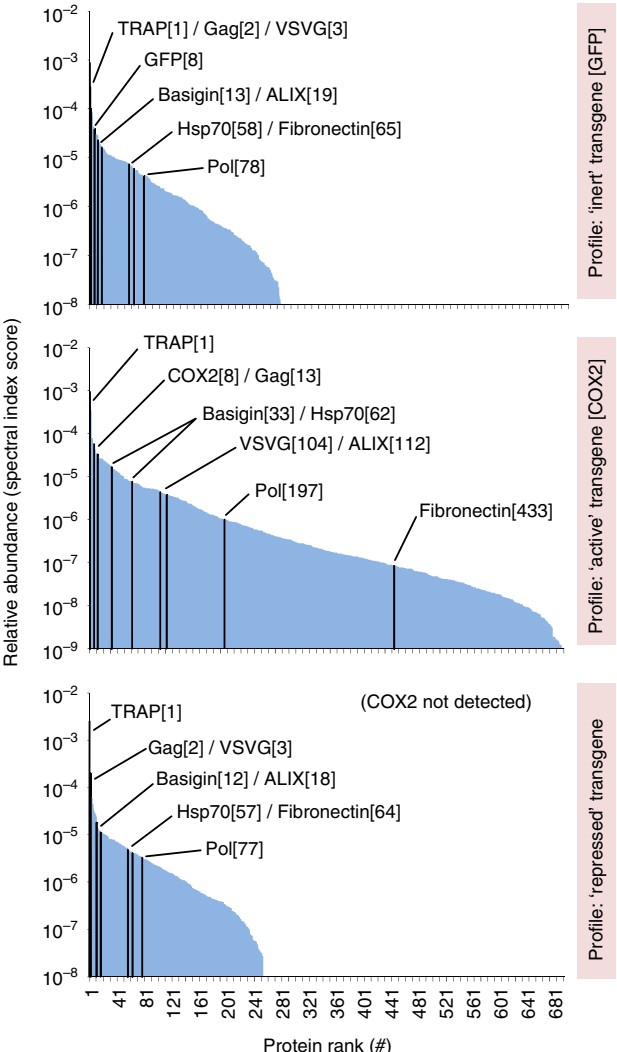

**Figure 5 | Protein profiles of serum-free EIAV-CMV-GFP, EIAV-CMV-COX2 and EIAV-CMV-tbsCOX2 vector preparations made in the presence of TRAP.** Quadruplicate serum-free vector concentrates were analysed by Mass Spectrometry Spectral Index-Normalized Quantitation (MS-SINQ). The profiles represent hits that varied by less than four-fold between replicate samples. EIAV-CMV-GFP and EIAV-CMV-COX2 represent 'inert' and 'active' transgene profiles respectively, compared to the 'repressed' transgene profile of EIAV-CMV-tbsCOX2. A large number of rare proteins were upregulated in EIAV-CMV-COX2 preparations (making up its profile 'tail'), and to a lesser extent also in EIAV-GFP. Changes within the top ~150 proteins (~90% of total detected) in EIAV-CMV-COX2 were less pronounced, with the exception of Gag, VSV-G and Fibronectin, which were reduced and appeared at lowing rankings. Common hits were used in three-way comparison of peptide pools of quadruplicates of each vector type, summarized in Table 1.

(Supplementary Fig. 5b). It is reasonable to suggest that these two effects of COX2 can be multiplied ($5 \times 50 = 250$-fold), which therefore largely explains the multiple log reduction in titres. We cannot exclude the possibility that down-/up-regulation of cellular factor(s) within EIAV-CMV-COX2 vector preparations may have contributed further to reduced virion activity (indeed, this is why we performed the MS-SINQ analysis to potentially identify such a factor). For instance, it is possible that the (85-fold) reduction in fibronectin within the EIAV-CMV-COX2 preparation could have been a contributing factor, which

**Table 1 | Examples of changes in protein abundance of EIAV vectors produced in the presence of 'inert' (GFP), 'active' (COX2) or 'repressed' transgene (COX-2) expression.**

| ID | Selected proteins | Fold-difference in relative abundance compared to EIAV-CMV-tbsCOX2 (spectral index) | | | | | |
| | | Experiment 1 (duplicates) | | | Experiment 2 (quadruplicates) | | |
| | | # | GFP | COX2 | # | GFP | COX2 |
|---|---|---|---|---|---|---|---|
| P19466 | TRAP | 1 | − 2.0 | − 3.4 | 1 | − 3.0 | − 2.9 |
| SPEIAV-19 | EIAV Gag polyprotein | 4 | 1.0 | − 6.1 | 2 | 1.4 | − 6.2 |
| P03522 | VSV-G (Indiana strain) | 7 | − 2.5 | − 27.4 | 3 | − 2.2 | − 48.3 |
| P62979 | Ubiquitin-40S | 15 | 1.3 | − 1.6 | 5 | 2.4 | − 1.5 |
| P35613 | Basigin | 70 | 2.4 | 3.2 | 12 | 2.7 | 2.0 |
| Q8WUM4 | ALIX | 28 | − 1.2 | − 3.9 | 18 | − 1.2 | − 4.9 |
| P07355 | Annexin A2 | 52 | 1.4 | − 1.7 | 34 | 1.9 | − 1.3 |
| Q00610 | Clathrin heavy chain 1 | 60 | − 1.1 | 2.4 | 42 | 1.3 | 1.6 |
| O00560 | Syntenin-1 | 81 | 1.1 | − 1.1 | 43 | − 1.1 | − 1.5 |
| P11142 | Heat shock cognate 71 kDa protein | 68 | 1.5 | 1.8 | 50 | 1.1 | 1.1 |
| P08107 | Heat shock 70 kDa protein | 87 | 1.5 | 2.6 | 57 | 1.7 | 1.7 |
| P04406 | Glyceraldehyde-3-phosphate dehydrogenase | 79 | 1.1 | 4.1 | 58 | 1.2 | 3.2 |
| P08133 | Annexin A6 | 138 | 3.3 | 4.5 | 60 | 3.8 | 2.8 |
| P02751 | Fibronectin | 352 | 1.0 | − 329.4 | 64 | − 1.3 | − 84.8 |
| Q08431 | Lactadherin | 97 | 2.1 | 1.4 | 67 | 1.9 | − 1.0 |
| P23528 | Cofilin-1 | 51 | − 1.9 | 1.4 | 69 | 1.4 | 2.3 |
| SPEIAV-19 | EIAV Pol polyprotein | 57 | − 1.2 | − 5.0 | 77 | − 1.1 | − 4.7 |
| P62158 | Calmodulin | 90 | 1.7 | 1.5 | 78 | − 1.4 | − 1.7 |
| P06733 | Alpha-enolase | 91 | 1.2 | 2.7 | 105 | 1.1 | 1.8 |
| P08238 | Heat shock protein 90-beta | 228 | 1.7 | 8.5 | 106 | 2.0 | 7.6 |
| Q09666 | Neuroblast differentiation-associated protein | 531 | 1.2 | − 28.1 | 154 | − 1.3 | − 186.1 |
| O00468 | Agrin | 330 | − 2.0 | − 24.9 | 206 | − 2.5 | − 88.3 |
| P35354 | Prostaglandin G/H synthase 2 (COX2) | 802 | nd | 2800 | na | nd | nc |

A selection of protein hits from the vector system and cellular proteins, in a three way SINQ comparison of common proteins (plus COX-2) within pooled peptides from EIAV-CMV-GFP and EIAV-CMV-COX2 vector preparations, compared to EIAV-CMV-tbsCOX2 vector preparations. Only hits that varied by less than four-fold between replicates are included. The data is presented from the first (duplicates) and second (quadruplicate) independent experiments, where Rank# and fold-difference of each hit relates to the same protein within the EIAV-CMV-tbsCOX2 profile in each independent experiment. The major component change in EIAV-CMV-COX2 was the > 2,800-fold increase in COX2 content. EIAV Gag, Pol and VSV-G were reduced in EIAV-CMV-COX2 vector preparations, which was in line with other measures of virion abundance of these vectors (Supplementary Fig. 5b). nd—not detected; na—not applicable; nc—not calculable.

has been reported to play a role in HIV-1 infection and can enhance retroviral vector transduction[27,28], although we have not formally pursued this hypothesis. However, these data show that the in-/direct effects of transgene protein during vector production are complex, and therefore the most pragmatic solution is to suppress transgene expression.

Surprisingly, TRAP was typically detected as the most abundant protein within vector virions regardless of the presence of a tbs in the vector genome, presumably due to its over-expression and passive incorporation during budding. Despite this, we were not able to detect an antibody response to TRAP (or COX-2) in rats injected intracamerally with purified EIAV-CMV-tbsCOX2, in contrast to VSV-G, to which 1/3 animals responded (Supplementary Fig. 6).

**Recovery of DNA-based vectors expressing an apoptotic gene.** To examine the utility of the TRiP system for other viral vector platforms we applied this approach to develop the TRiPAdeno (Fig. 6a) and TRiPAAV (Supplementary Fig. 8a) systems. We first designed an Adenovirus type 5 shuttle plasmid containing the CMV-tbsGFP cassette (Fig. 6b), and assessed the repression of GFP during co-transfection of HEK293T cells with Ad5 packaging backbone using the RAPAd in-mammalian cell recombination system. Approximately 100-fold repression of GFP transgene expression was observed during first generation vector isolation (Fig. 6c), as well as 2–3 orders of magnitude of transgene repression during a single round of amplification in cells stably expressing TRAP (HEK293T.TRiP) (Fig. 6d). These initial results encouraged us to demonstrate proof-of-principle of the utility of the TRiP-Adeno system, by attempting to produce adenoviral vectors

expressing the pro-apoptotic protein Bcl-2-associated X protein (Bax). The shuttle plasmid pAdShuttle-CMV-tbsBax-iGFP was constructed so that GFP could be used as a marker for vector titration (Fig. 6b). While performing the recombination step in HEK293T and HEK293T.TRiP cells to make Adeno-CMV-tbsGFP or Adeno-CMV-tbsBax-iGFP vectors, we observed that only the HEK293T.TRiP cells supported expression of GFP from pAd-Shuttle-CMV-tbsBax-iGFP, indicating that expression of Bax within HEK293T cells produced a strong inhibitory effect on global gene expression, presumably as a consequence of induction of apoptosis (Supplementary Fig. 7a). It should be noted therefore that cells transduced with the Adeno-CMV-tbsGFP and Adeno-CMV-tbsBax-iGFP vectors have opposite GFP phenotypes depending on whether TRAP is co-expressed. We were only able to observe cytopathic effect (CPE) within the HEK293T.TRiP cells (for either vector type) during the 14 day recombination phase, perhaps reflecting some intrinsic characteristic of the TRiP cell line over its parent. Crucially, we observed that the plaques that formed within cultures producing Adeno-CMV-tbsBax-iGFP had a higher degree of GFP expression compared to the cells surrounding them (Supplementary Fig. 7b), indicating that amplification of GFP-expressing vector had occurred.

We therefore used the vector stocks of both vector types generated in HEK293T.TRiP cells to model vector amplification from material that might otherwise be generated in bacteria or in vitro, where the toxic effect of Bax would not have been present. We took each of the triplicate vector stocks and initiated four rounds of vector amplification on either HEK293T or HEK293T.TRiP cells. In the first round we observed massive levels of cell-rounding and detachment of HEK293T cells post-transduction with Adeno-CMV-tbsBax-iGFP, and these cells

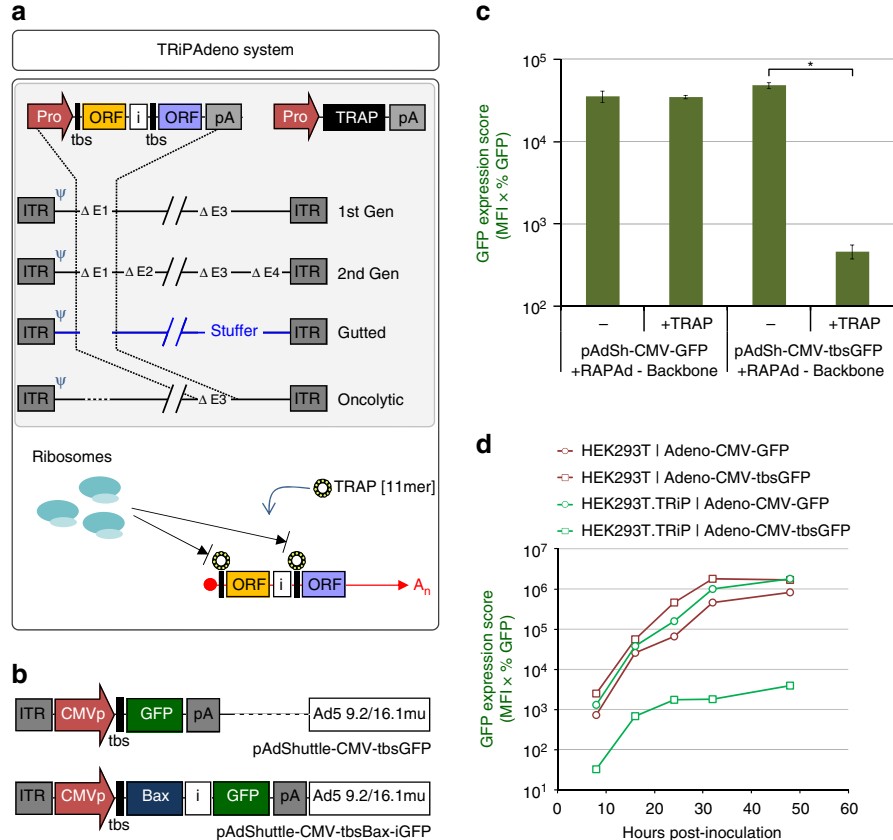

**Figure 6 | Repression of GFP during production of TRiPAdeno-GFP vectors.** (**a**) The TRiPAdeno system, which requires: vector genome (with tbs-modified transgene(s)), helper functions and a TRAP expression cassette. TRiPAdeno is suited to the production of first/second generation Ad vectors, as well as Gutted (Helper-dependent) and Helper-independent Oncolytic Ad vectors. (**b**) The tbs-modified adenoviral type 5 shuttle plasmids used in the study, containing the 'left' ITR, transgene cassette and adenovirus homology block of 9.2–16.1 map units. Bax was used to exemplify production of a vector expressing high-levels of a toxic gene (see text and Fig. 7). (**c**) Repression of GFP expression during vector recombination together with the pRAPAd back-bone plasmid in HEK293T cells. Data are mean average values ± s.d. [$\log_{10}$-transformed data] ($n = 3$); *$P < 6.7 \times 10^{-5}$ [Welch's $t$-test]. (**d**) Vector stocks generated from (**c**) were used to transduce either HEK293T or HEK293T.TRiP cells at a multiplicity of infection (MOI) of 0.01, and replicate cultures analysed by flow cytometry at different points post-inoculation during one round of vector amplification phase to generate GFP Expression Scores. (Pro, promoter; ITR, inverted terminal repeat; Ψ, packaging signal; tbs, TRAP-binding sequence; ORF, open-reading frame; 'i', internal ribosomal entry site (encephalomyocarditis virus); polyA, polyadenylation signal; An, poly-adenides).

appeared to have negligible levels of GFP fluorescence (Fig. 7a). In contrast, HEK293T.TRiP cells were GFP-positive and gave an appearance consistent with Adenovirus-related CPE. Each subsequent amplification round was initiated with <1% of vector material from the preceding round.

Crude vector stocks from amplification phase IV were titrated on both HeLa and HeLa-TRAP cells to allow GFP TU per ml titre values to be generated for Adeno-CMV-tbsGFP and Adeno-CMV-tbsBax-iGFP vectors, respectively (Fig. 7b). Strikingly, HEK293T.-TRiP cells supported amplification of Adeno-CMV-tbsBax-iGFP vector to titres in excess of $1 \times 10^8$ GFP TU per ml, which exceeded the Adeno-CMV-tbsGFP titres produced from these cells. HeLa-TRAP cells transduced with material derived from amplification of Adeno-CMV-tbsBax-iGFP vector on HEK293T cells did not produce any quantifiable titre. The apparent titre of HEK293T.-TRiP-produced Adeno-CMV-tbsBax-iGFP vector on HeLa cells ($1 \times 10^6$ GFP TU per ml) is believed to reflect the initial burst of expression of both transgenes before the effects of global protein expression inhibition by Bax; this effect appeared to be less potent in HeLa cells compared to HEK293T cells, suggesting that the latter may be more sensitive to Bax over-expression.

We performed qPCR on purified vDNA using a GFP primer/probe set (Fig. 7c), which showed that Adeno-CMV-tbsBax-iGFP

vectors produced in HEK293T.TRiP cells had crude particle titres of $3 \times 10^{10}$ genomes per ml, which represented a >150,000-fold excess over signal detected in material derived from HEK293T cells. Since we were able to detect GFP sequences in material derived from HEK293T cells above cell background, this may have represented the early stages of amplification of a recombinant form of Adeno-CMV-tbsBax-iGFP that presumably had lost both GFP expression (since we observed none) and/or functional Bax expression, although we did not observe any appreciable CPE in these cultures in any round of amplification beyond round 1.

To assess the stability and functionality of the transgene cassettes within the amplified vector stocks, we transduced both HeLa and HeLa-TRAP cells with vectors at a multiplicity of infection (MOI) of ~5, based on the biological GFP titres of these stocks. Since we were not able to detect any biological titre in Adeno-CMV-tbsBax-iGFP vector material derived from HEK293T cells, we transduced cells with as much material as practicable, which resulted in a ~15-fold volume excess over Adeno-CMV-tbsBax-iGFP vector material derived from HEK293T.TRiP cells. We analysed post-transduced cell lysates for levels of GFP, Bax and cleaved Poly (ADP-ribose) polymerase (clv-PARP; a marker of apoptosis) by SDS-PAGE and immuno-blotting (Fig. 7d). This key experiment demonstrated that: [1] the

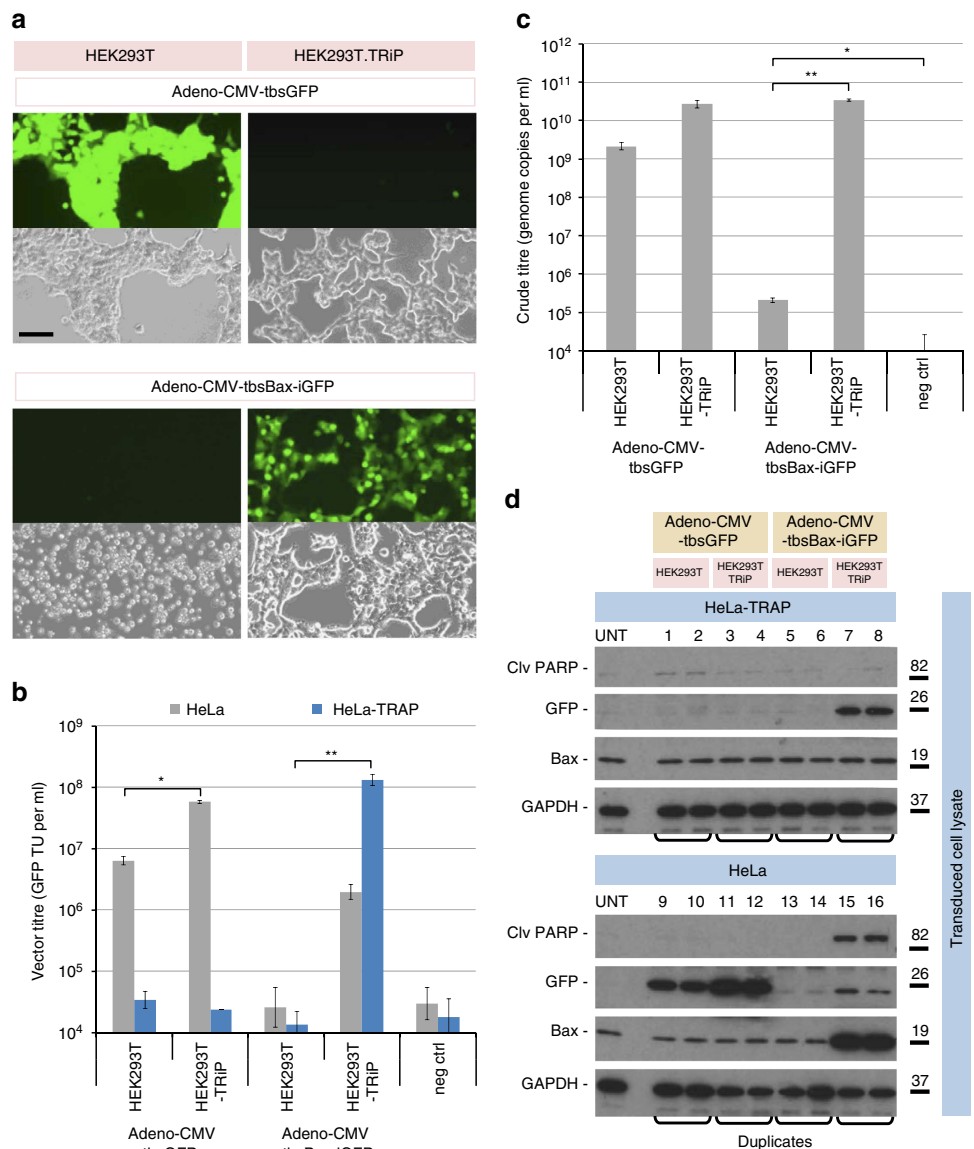

**Figure 7 | Proof-of-principle TRiPAdeno vector production using the apoptotic transgene Bax.** (**a**) Representative phase or fluorescent microscopy images of HEK293T or HEK293T.TRiP cell cultures in the first amplification phase post-inoculation with Adeno-CMV-tbsGFP or Adeno-CMV-tbsBax-iGFP vectors produced in HEK293T.TRiP cells (black scale bar = 100 μm). (**b**) Vector stocks from the final round of amplification/passaging were titrated on HeLa cells and HeLa-TRAP cells. Data are mean average values ± s.d. [$\log_{10}$-transformed data] ($n = 3$); *$P < 5.4 \times 10^{-4}$, **$P < 1.5 \times 10^{-4}$ [Welch's $t$-test]. (**c**) Vector stocks from the final round of amplification/passaging were titrated by qPCR of purified vDNA using a GFP primer/probe set. Data are mean average values ± s.d. [$\log_{10}$-transformed data] ($n = 3$); *$P < 3.1 \times 10^{-2}$, **$P < 6.7 \times 10^{-8}$ [Welch's $t$-test]. (**d**) Vector stocks from the final round of amplification/passaging were used to transduce HeLa and HeLa-TRAP cells at an MOI ~5, and transduced cell lysates analysed for functional expression of transgene cassettes by immunoblotting to Cleaved-PARP (induced by Bax over-expression), Bax, GFP and GAPDH (molecular weight marker in kilodaltons). Triplicate Adenoviral vector stocks were derived and amplified/passaged from three independent recombination steps; qPCR and immuoblotting were each repeated twice.

'tbs controlled' ORFs of GFP and Bax in Adeno-CMV-tbsGFP and Adeno-CMV-tbsBax-iGFP respectively, had maintained their ability to be repressed by TRAP (Fig. 7d: see GFP panels, lanes 1–4 [OFF] versus 9–12 [ON] and Bax panels, lanes 7–8 [OFF] versus lane 15–16 [ON]; [2] high-level Bax expression (above endogenous) was only detected in HeLa cells transduced with Adeno-CMV-tbsBax-iGFP vector made using the TRiP system (Fig. 7d: see Bax panels, lanes 13–14 [HEK293T] versus 15–16 [HEK293T.TRiP]); and [3] only the high-level of Bax expression led to induction of cleaved PARP (Fig. 7d: see clv-PARP and Bax panel; lanes 15–16), demonstrating that the Bax protein that was expressed was also functional.

Finally we evaluated the TRiPAAV system (Supplementary Fig. 8a) versus the standard approach in producing self-complementary (sc)AAV2-GFP vectors by co-transfecting HEK293T cells with packaging/helper plasmids with either pscAAV2-CMV-GFP or pscAAV2-CMV-tbsGFP with or without TRAP plasmid. GFP Expression scores were generated from replicate production cell cultures after flow cytometry, and purified vectors were titrated on HEPG2 cells (Supplementary Fig. 8b). These data show that 100-fold repression of the transgene occurred during AAV vector production and the TRAP-tbs configuration does not intrinsically affect vector activity, as the biological titres of TRiPAAV-produced vectors

were equivalent to those made by the standard method. We were then able to recover titres of scAAV2-CMV-tbsBax vectors to the same level as GFP-expressing vectors in HEK293T cells, only when TRAP plasmid was supplied in during vector production (Supplementary Fig. 8c).

## Discussion

It is widely known that viral vector-encoded transgene expression can adversely affect vector titres. This effect may be anticipated from the outset of vector development, particularly when utilizing known cytotoxic transgenes, for example, in cancer-killing vectors[10,29] or anti-HIV-1 genes that encode restriction factors that affect retroviral vectors, for example, APOBEC3G (ref. 30) and TRIMcyp (ref. 31). It is conceivable that viral vector genomes expressing gene-editing transgenes may also be cytotoxic depending on the degree/type of off/on-target effects, particularly in the context of stable producer cell lines (PrCLs). The effects of other transgene proteins may be less predictable or more subtle, for example by affecting vector packaging or exclusion of glycoproteins from the virion envelope. Alternatively, cellular proteins may be incorporated into or excluded from virions, which may play a key role in virion activity[25,26,32]. To address this issue we have devised the TRiP system to minimize vector transgene expression in production cells and thus reduce any such effects.

We initially identified COX2 expression to be highly detrimental to EIAV vector titres. However, upon the application of TRiP system to the production of these vectors we observed up to a 600-fold recovery in titres, demonstrating the potency of this method of transgene repression. Although the underlying mechanism through which COX2 expression reduces titres is unclear we did observe that it correlated with a ∼50-fold reduction of VSV-G incorporation into vector virions. We have previously shown that the degree of VSV-G incorporation into lentiviral vector virions modulates their activity[33]. Similarly, expression of Factor-VIII has been shown to have a similar effect on EIAV-based lentiviral vectors[14]. However, we cannot rule out a contribution of cellular factor(s) (up or down-regulated) in the reduction of titres of EIAV-CMV-COX2 vector preparations. COX-2 is a multifunctional protein, and its expression is variably altered by different viruses during infection[34]. In this study, the number of cellular proteins in the top 250 proteins up- or down-regulated by more than 10-fold within EIAV-CMV-COX2 vector preparations were 24 and 14, respectively. While some of these were glycoproteins involved in basement membrane or extracellular matrix formation (and it is tempting to link this to the reported role of cell migration by COX2 in immortalized cells[35]), we cannot exclude the possibility that mere over-expression of COX2 (itself being a glycoprotein) simply impeded the secretory pathway or induced endoplasmic reticulum stress, with some glycoproteins being more sensitive than others. This perhaps reflects the inherent as well as unpredictable biological activities of therapeutic transgene proteins in mammalian cells, and justifies the need to prevent them during vector production.

The potential impact of transgene expression in production cells on vector titres and strategies to address the issue has been the focus of previous studies. In contrast to the translational block imposed by the TRiP system described here, others have attempted to suppress vector transgene expression by utilizing transcriptional control[29,36,37] or through use of a ligand-induced riboswitch[38]. Employment of transcriptional repressor systems may be limited in the magnitude of repression due to upstream read-through from the 5′LTR during generation of full length packageable RNA (for example, retroviral vectors), or by

ITR-driven genome replication (for example, AAV/Adeno vectors). It also may be difficult to engineer the repressor binding site within the desired transgene promoter without affecting transcriptional start site usage, while also maintaining repression levels. The presence of highly structured riboswitches within transgene UTRs may result in attenuated transgene expression in target cells, such as via Protein Kinase R activation[39]. Furthermore the use of riboswitches in lentiviral or retroviral vectors would lead to cleavage of the vector genomic RNA, with a likely adverse effect on vector titres.

One attractive approach is the strategy of utilizing the RT step during retroviral vector transduction to activate/up-regulate the transgene cassette only in target cells—first termed 'ReCon' vectors[40–43]. While these approaches have variable success in the degree of transgene repression during vector production, it is clear that the generation of the vector genome molecule can be impacted. This approach is typically troubled by reduced titres of upwards of 2- to 10-fold compared to the benchmark GFP standard, even before leaky expression of a toxic transgene is taken into account. This is likely due to unwanted side effects of the types of modifications employed. Those studies indicate that insertion of (inverted) heterologous transcription elements within LTR regions can lead to promoter competition or perhaps interfere with aspects of reverse transcription. It is conceivable that inversion of the transgenic cassette may result in new splice sites or other unstable elements, leading to a reduction in the amount of vRNA available for packaging.

The mechanistic advantage of the TRiP system is that translational repression does not intrinsically impede the DNA- or RNA-based viral vector systems tested empirically here, and can repress cap-dependent and -independent protein translation levels of multiple transgenes by over two orders of magnitude. This is mediated via a relatively short 55 nucleotide sequence, which does not significantly impact on vector capacity, and works independently of the promoter employed. Surprisingly, reverse transcription of lentiviral vectors was unaffected by the presence of a tbs within the genome, suggesting that the TRAP detected within virions was either not bound to vRNA or that reverse transcriptase can pass through the TRAP-tbs complex. The degeneracy of the tbs consensus allows for flexibility in the final sequence utilized; obviously ATG codons and cryptic splice sites should be avoided. The TRAP protein is not toxic to production cells, and so if the protein is able to interact with cellular RNAs, these are either relatively weak associations or stronger interactions do not result in any cytotoxic phenotype. We therefore expect the TRiP system to be broadly applicable to most, if not all viral vector platforms.

The lack of antibody response to TRAP in rats treated intracamerally with EIAV-CMV-tbsCOX2 in the eye (an organ/route known to be particularly sensitive to inflammation) provides evidence that the presence of TRAP protein within retroviral vector virions is unlikely to be problematic. In support of this, it is our experience in the clinic that human antibody responses to protein in purified VSV-G pseudotyped lentiviral vector preparations were only found at the highest doses, and were primarily to the viral glycoprotein, which is present on the virion surface[44]. Nevertheless, future investigation into potential pre-existing anti-TRAP responses in human sera may be warranted for application of the TRiPRetro/Lenti platform.

Production titres of therapeutic vectors can be increased by several orders of magnitude using the TRiP system, and in some cases at the platform benchmark. The ability of the TRiPAdeno and TRiPAAV systems to produce both Bax-expressing and GFP-expressing vectors to equivalent titres is particularly impressive, and to our knowledge the first time this has been

achieved using a powerful, constitutive promoter. This may genuinely offer the gene therapy/vaccine field the opportunity to exploit a new category of viral vectors expressing high-levels of potently toxic transgenes, and broadens the potential repertoire of therapeutic proteins that can be evaluated and delivered by viral vectors.

We are exploring other potential benefits; it is conceivable that the incorporation/association of the transgene protein in/on virions may affect downstream processing such as virion capture or filtration, particularly if the transgene protein is incorporated into the virion surface. While some transgene proteins may be considered inert (such as GFP) —with low/no impact on crude titres—the effect on the protein profile of vector preparations and presumably of the cells from which they are derived is not likely to be zero. This has implications for PrCL development, where transgene protein may alter their growth characteristics compared to the packaging cells from which they are derived, perhaps making the development path less predictable. In addition, there is potential to present a varying identity/quantity of cellular, as well as transgene protein, to the immune system during direct *in vivo* gene therapy.

Finally, a major advantage of the TRiP system is the standardization of manufacture within a bio-similar vector platform since only the vector components and TRAP will be expressed in production cells. As more gene therapies succeed in the clinic, it is likely that regulators will place greater emphasis on robust generation of purer vector product and more stringent product specifications. The TRiP system will therefore have broad appeal in the viral vector community and its simple application has the capability of transforming gene therapy approaches that may currently be derailed by poor vector production, and perhaps opens the door to the production of a new variety of viral vectors expressing potently toxic proteins, such as armed-oncolytic vectors. The TRiP system will be a useful platform in the clinical development of viral vectors expressing problematic transgenes and is a major step towards the generation of vector production systems that are 'independent' of the transgene cassette.

## Methods

**Plasmids.** The TRAP expression plasmid pEF1α-coTRAP[H6] encodes *Homo sapiens*-codon optimized *Bacillus subtilis* TRAP [GenBank: ABN13202.1] fused with a C-terminal 6 × histidine tag. The stable TRAP expression cassette used to make HEK293T.TRiP cells additionally encoded the EMCV IRES and either the blasticidin (HEK293T.TRiP[3D]) or puromycin (HEK293T.TRiP[H1]) resistance genes. The GFP reporter plasmids encode enhanced GFP from Clontech (Takara) and the wild type EIAV LTR (nt1-323 of SPEIAV-19), which contains a polyadenylation signal. The 5′UTRs were either 85 nt from CMV promoter (control) or a UTR-tbs sequence (*tbs*; modified from ref. 20), ORF start):

5′GGTCAGATCCGCTAGCGCTACCGGACTCAGATCTCG*AGTTTAGC GGGTGGAGAAGAGCGGAGCCGAGCCTAGCAGAGACGAGAAGAGC*TAC AGCCACCATG. For other promoter-driven tbsGFP reporters, the tbs-kozak sequence above was inserted directly between the existing UTR of the respective promoter and the ATG codon.

EIAV-based vector genomes were derivatives of pONY8.9NCZ (ref. 14), wherein the lacZ gene was replaced with either eGFP (EIAV-[ ± tbs]GFP), codon-optimized COX-2 [NCBI ref seq: NP_000954.1] (EIAV-[ ± tbs]COX2) or Factor VIII (ref. 14) (EIAV-[ ± tbs]Factor VIII). The EIAV GagPol and VSV-G expression plasmids have been described previously[33,45]. Our HIV-1-based vector platform has been described previously[4,46,47]. The CAR5T4 codon-optimized transgene encodes a chimeric antigen receptor targeting the 5T4 cancer antigen[48]. The 5′UTRs within transgene cassettes were those stated for the GFP reporter plasmids unless otherwise stated.

The scAAV vector genomes contained a trs deletion and SV40 polyadenylation signal, and were fully re-derived by GeneArt/Life Technologies. The 5′UTRs within transgene cassettes were those stated for the GFP reporter plasmids unless otherwise stated.

The first generation Adenoviral shuttle vector plasmid pAdShuttle-CMV-GFP was pacAd5-CMV-GFP(Control) from the RAPAd system (CellBiolabs), and the 5′UTR-tbs sequence cloned into this plasmid to generate pAdShuttle-CMV-tbsGFP. The pRapAd-backbone plasmid was pacAd5 9.2–100 from the same system.

The Bax open reading frame (GeneBank: AAH14175.1) was codon-optimized (*Homo sapiens*) by GeneArt/Life Technologies, and cloned into the AAV or Adeno vector genome/shuttle plasmids.

**Cell lines.** Both the HEK293T.TRiP[3D] (used for all indicated EIAV vector studies) and the HEK293T.TRiP[H1] (used for Adenoviral and AAV vector studies) cell lines were derived from a GMP bank of HEK293T cells; all have subsequently been verified by PCR single-locus technology (Eurofins Medigenomix Forensik GmbH) and verified against the DSMZ database. HeLa cells (ECACC, 93021013) were transduced with MLV-coTRAP[H6]-iresPuro and selected with puromycin to generate a pool resulting in 'HeLa-TRAP' cells; these were used for biological titration so that no further replication of Ad vectors could occur (HeLa cells being E1 deficient), unlike HEK293T/HEK293T.TRiP cells. HEK293T-based cell lines were tested negative for mycoplasma.

**Lentiviral vector production.** The standard scale production of lentiviral vectors was in 10 cm dishes under the following conditions (all conditions were scaled by area when performed in other formats): HEK293T cells were seeded at $3.5 \times 10^6$ cell per plate in 10 ml complete HEK293T media (DMEM (Sigma) containing 10% heat-inactivated FBS (Gibco), 2 mM L-glutamine (Sigma), 1 × NEAA (Sigma)) and incubated at 37 °C in 5% CO$_2$ through-out production. Approximately 24 h later, cells were transfected using the following mass ratios of plasmids for EIAV vector production: 4 μg genome, 2 μg gag/pol, 0.1 μg VSV-G, and 0.56 μg TRAP or pBlueScript. This represented a genome:TRAP plasmid molar ratio of 5:1. Transfection was mediated by mixing DNA with Lipofectamine 2000CD in Opti-MEM according to manufacturer's protocol (Life Technologies). Sodium butyrate (Sigma) was added ∼18 h later to 10 mM final concentration for 5–6 h, before 10 ml fresh complete media replaced the transfection media. For serum-free vector production for SINQ analysis, vector was harvested in Freestyle-293 media containing 0.1% Cholesterol lipid (Life Technologies). A single harvest was performed 20–24 h later, vector supernatant filtered (0.22 μm) and frozen at − 20/ − 80 °C. For HIV-1 vector production, the following mass ratios of plasmids were used per plate: 4.5 μg genome, 1.5 μg gag/pol, 1.1 μg rev, 0.7 μg VSV-G and 1 μg TRAP or pBlueScript. This represented a genome:TRAP plasmid molar ratio of 5:2. The remainder of the process followed that stated for EIAV vectors. Note that for GFP and luciferase/GFP reporter plasmid evaluations, the GFP:TRAP plasmid molar ratio was 5:2, and pBluescript was used to ensure total DNA transfected was the same in all conditions.

**Adenoviral vector production.** Production of first generation GFP expressing vectors occurred by recombination of pAdShuttle-CMV-[ ± ]tbsGFP plasmids with the pRapAd backbone in HEK293T cells. HEK293T cells were seeded at $5.8 \times 10^5$ cells per 6-well in 3 ml complete media, and incubated at 37 °C in 5% CO$_2$ through-out production. Approximately 20 h later, cells were transfected using the following mass ratios of plasmids for Adeno vector production: 166 ng pRapAd-backbone, 666 ng shuttle, and 666 ng TRAP or pBlueScript. This represented a shuttle:TRAP plasmid molar ratio of 5:2. Transfection was mediated by mixing DNA with FuGENE 6 (2.9 μl per μg DNA) and Opti-MEM (112 μl per μg DNA) according to manufacturer's protocol (Roche). Approximately 20 h later, 3 ml fresh complete media replaced the transfection media and cultured until ∼50% cpe was observed and cells harvested. Cells were freeze-thawed three times before clarification and filtration (0.22 μm) of crude vector stocks. Amplification of vectors from crude stocks was carried out by infection of HEK293T or HEK293T.TRiP cells at MOI 0.01 and cultures incubated for 48 h. Replicate cultures were sacrificed for GFP analysis at stated times during amplification.

Amplification of Adeno-CMV-tbsGFP and Adeno-CMV-tbsBax-iGFP vector stocks occurred by inoculation of 6-well cultures of HEK293T or HEK293T.TRiP cells pre-seeded with $9 \times 10^5$ cells per well, with 40% of material generated from the freeze-thaw phase of HEK293T.TRiP cell cultures that displayed cpe. For amplification phases 2–4, fresh cell cultures at 10 cm$^2$ plate scale (pre-seeded 20 h prior with $4.5 \times 10^6$ cells per plate) were inoculated with ∼0.7% of freeze-thawed/clarified material from the preceding round in 10 ml media; this generated ∼1 ml of crude vector in serum-free media at each round.

**Assays for measuring transgene expression.** Expression of GFP was carried out by flow cytometry (FACSVerse, BD Biosciences), out-gating dead cells beforeanalysis of FL1 channel events. GFP Expression scores were calculated by multiplying percent GFP-positive cells by the median fluorescence intensity of events within the GFP-positive gate. GFP, COX-2, Bax, Cleaved-PARP and GAPDH expression in cell lysates was carried out by SDS–Polyacrylamide gel electrophoresis (SDS–PAGE) under reducing conditions, Western transfer and immunoblotting using antibodies to GFP (ab290, Abcam), COX-2 (CAY160112, Cayman Chemical), Bax (ab32503, Abcam), Cleaved-PARP (ab140360, Abcam) and GAPDH (ab9485, Abcam) Species-specific HRP conjugated secondary antibodies were used at 1:1,000 dilution. Expression of luciferase was carried out using the Dual luciferase reporter assay (Promega). Full images of cropped blots are presented in Supplementary Figs 9 and 10.

**Detection of cytoplasmic GFP RNA copy-number by qRT-PCR.** Total cytoplasmic RNA was extracted and purified from cells using an RNeasy mini kit (QIAGEN). One microgram of RNA was treated with DNAse I (Ambion) for 1 h before inactivation. 50 ng of DNAase I-treated RNA was used in qRT-PCR reactions comprised of Taqman One step RT-PCR master mix (Life Technologies) under standard chemistry RT-PCR cycling conditions using a QuantStudio 6 (Life Technologies). A primer-FAM-probe set was designed to specifically detect a region of the GFP ORF. Negative control reactions contained no RT to control for DNA contamination.

**Viral vector titration assays.** DNA integration assay for titration of lentiviral vectors has been described elsewhere[44]. Titration of GFP-encoding viral vectors occurred by serial dilution and 5–6 h transduction of target cells (in FBS- and $8 \mu g \, ml^{-1}$ polybrene-containing media for lentiviral vectors, and in serum-free media for Adenoviral-based vectors) before addition of fresh media, followed by incubation for 2 days. Target cell counts were made before transduction. Cultures were analysed for percent GFP expression using a FACSVerse and vector titres calculated accordingly. For qPCR of Adeno vectors, vDNA was extracted and purified from 90 μl of crude vector material using the Qiagen DNA minikit using carrier RNA, generating 60 μl of pure DNA. Neat or 10-fold diluted DNA served as template for TaqMan qPCR reactions using Taqman Universal PCR master mix (Life Technologies). Reactions were performed under standard chemistry PCR cycling conditions using a QuantStudio 6 (Life Technologies). A primer-FAM-probe set was designed to specifically detect a region of the GFP ORF:
Fwd - 5′ · CAACAGCCACAACGTCTATATCATG · 3′,
Probe - 5′ · [FAM]CCGACAAGCAGAAGAACGGCATCAA[TAMRA] · 3′,
Rev - 5′ · ATGTTGTGGCGGATCTTGAAG · 3′

**Protein analysis of EIAV-based vectors by SINQ.** Quantification was carried out by Cytome Technologies (Upper Heyford, UK) as follows: purified protein mixtures were denatured and trypsinized using the FASP protocol to ensure efficient digestions and optimal recovery[49]. Peptide digests were analysed using a QExactive mass spectrometer (Thermo Scientific). Proteins were identified using the Trans Proteomics Pipeline and relative protein abundance was measured by label-free quantitative mass spectrometry using the Normalized Spectral Index SIN (ref. 50). The three-way comparison was carried out by using pooled data from the results of the individual analyses of each duplicate or quadruplicate vector preparation. The initial set of common proteins were retrospectively filtered by removing hits that varied by more than four-fold between replicate sample analyses of each vector type. The COX-2 protein data were manually included in the final ranking list, because it was not a common protein (not present in EIAV-GFP preparations).

**Statistical analysis.** Statistical analysis was performed by Welch's unequal variances $t$-test (two-tailed, type 3) using $\log_{10}$-transformed data. $P < 0.05$ was considered to be significant.

**Data availability.** The authors declare that the data supporting the findings of this study are included within the article and its Supplementary Information file, or are available from the authors on request, subject to a confidentiality agreement.

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

## Acknowledgements

The authors would like to thank the team at Cytome Technologies for their work and helpful suggestions regarding the SINQ analysis.

## Author contributions

D.C.F. conceived and directed the project (except the animal study of Supplementary Fig. 6, which was directed by S.E.), wrote the manuscript and created display items, as well as generated the data in Figs 2b,c and 5 and Table 1, Fig. 6e,f, and Supplementary Figs 5, 7, 8b. H.E.M. generated the data in Figs 4b and 6c,d, and Supplementary Fig. 4a–c. J.W. generated the data in Figs 2d and 3, and Supplementary Figs 1 and 2. B.R.K. developed the HeLa-TRAP cells, and generated the data in Fig. 6g and Supplementary Fig. 8c. C.R.V. generated the data in Supplementary Fig. 8h. H.E.M., S.T. and T.T.M. developed the HEK293T.TRiP cell lines. V.K. generated the data in Fig. 4d,e, and contributed equally with S.I. to generate Supplementary Fig. 6. S.I. also generated the data in Supplementary Fig. 3, and contributed to Supplementary Fig. 4d together with G.D. Y.L. generated the data in Fig. 4c. N.G.C. advised on SINQ analysis. K.A.M. advised and contributed to manuscript revisions.

## Additional information

**Competing interests:** The work described was fully funded by Oxford BioMedica Ltd., and all authors at the time of submission, except T.T.M. and S.T., were employees and hold stock or stock options within the company.

**Publisher's note**: 

