## [Peer Review File · Nature Communications]

Reviewers' comments:

Reviewer #1 (Remarks to the Author):

The manuscript by Maunder et al. describes the use of the Tryptophan RNA-binding Attenuation Protein (TRAP) responsive element in gene transfer vector transgene cassettes that negatively impact on viral production due to presumed cell toxicity during the production. This observation has been a longstanding issue in the manufacturing at scale of gene therapy vectors with only suboptimal solutions (e.g. altering promoter). The approach outlined here is novel and the data demonstrating its utility in EIAV lentiviral vectors robust. Inclusion of the TRAP in the 5' untranslated region of a transgene construct only adds minimal coding and does not appear to interfere with lentiviral genome processing. The mechanism of attenuation in this context appears to be largely at the level of protein translation. The findings in this study are of interest to the field of gene therapy and may provide a creative workaround to a longstanding problem in the field. Data is provided for EIAV lentiviral vectors which likely extends to the broader class of retroviruses. Data for demonstrating the utility and benefit other vector systems such as AAV and Ad is not provided, although using a non-toxic GFP molecule experiments indicate a significant attenuation of protein expression in producer cells and no overall effect of including TRAP-responsive sequences in these vector systems.

While the main findings of the paper are of interest and consistent with the conclusions, there are several issues we would like to see addressed.

- 1- The abstract provides too little information on the approach taken and the scope of the results.
- 2- the manuscript would benefit from a more extensive, clear description of the bacterial TRAP mechanism and biological role, possibly with a schematic as a panel in Fig 1.
- 3- The authors measure vector titer by equal volume transduction, then quantifying integration events, and show that there are dramatically fewer integration events in the CMV-toxic transgene (EIAV-COX2) context than when the TRAP/tbs are included in the transfected production plasmids (EIAV-tbsCOX2). Fig 4 indicates that there appears to be 5-7 fold more viral proteins present in the EIAV-tbsCOX2 preps than normal pEIAV-COX2 produced vectors, what is also demonstrated is that the proteomic profiles of each prep are significantly different. It is not clear if variations in integration events are solely attributable to a reduced number of virions present, which the authors claim is the case, or if transduction efficiency is affected by the vector composition.
- 4- the MS-SINQ method requires a minimum of quadruplicate samples for sufficient confidence in the results, whereas the present work only uses duplicates.
- 5- The effects of vector preps with different protein contents on target cells are difficult to interpret, and interpretation of the data would benefit from an additional physical titer titration.
- 6- While the data provided on AAV and Ad does suggest that the relevance of this system can be extended to other vector systems, no demonstration of its benefit with a toxic transgene is provided. The manuscript would substantially increase in impact if indeed these data would demonstrate its utility across the most commonly used gene transfer vector systems. In absence of these data, we would recommend to include the present data however do not refer to those at 'proof-of-concept' in the abstract.

Reviewer #2 (Remarks to the Author):

Maunder and colleagues describe an elegant new approach to mitigate the negative impact of transgene expression in the efficiency of recombinant vector production. This bacterial translation repression system requires insertion of a small 55 nucleotide sequence (tbs) in the transcriptional cassette. The work is excellent demonstrating the effectiveness of this approach to increase lentivirus vector titers by 1-2 orders of magnitude, which is an important development for broad implementation of some of the promising ex vivo lenti-based gene therapies. This approach is also

shown to be effective in repressing Cap-dependent and independent expression from bicistronic lentivirus vectors. Moreover the major changes in the proteomic content of a tbs containing lentivirus vector produced with and without TRAP are at the level of either the transgene itself or viral proteins, although fibronectin levels also appear to be higher in the presence of TRAP which may explain in part the higher transduction titers. This is an important advancement in the field of recombinant viral vector production. The data on transgene repression during adenovirus or AAV production is solid but the usefulness of this approach remains to be demonstrated in these systems which have inherently different mechanisms and limitations. It is important to consider that at least in the analyzed lentivirus/transgenes the major effect appeared to be at the level of VSV-G incorporation into the lentivirus virion.

Minor considerations:

1. It would be interesting to determine whether the overall level of VSV-G in producer cells changes or the effect of transgenes is at the level of its incorporation into virions. Same for virion proteins.
2. At least in some preps in addition to the functional titer, the authors should also analyze the particle content by a simple p24 ELISA as it would clarify whether the decreased titer could also be caused by inefficient packaging of the vector genome, but not a direct effect on virion assembly.
3. These details do not change the enthusiasm for this manuscript.

This work is considered high impact, but in this reviewer's opinion it is better suited for a specialty journal focusing on gene therapy methods. Nonetheless congratulations on an elegant system to address the production limitations that are known in the field.

Response to reviewer's comments: manuscript NCOMMS-16-02222(A)

Reviewers' comments:

Reviewer #1 (Remarks to the Author):

The manuscript by Maunder et al. describes the use of the Tryptophan RNA-binding Attenuation Protein (TRAP) responsive element in gene transfer vector transgene cassettes that negatively impact on viral production due to presumed cell toxicity during the production. This observation has been a longstanding issue in the manufacturing at scale of gene therapy vectors with only suboptimal solutions (e.g. altering promoter). The approach outlined here is novel and the data demonstrating its utility in EIAV lentiviral vectors robust. Inclusion of the TRAP in the 5' untranslated region of a transgene construct only adds minimal coding and does not appear to interfere with lentiviral genome processing. The mechanism of attenuation in this context appears to be largely at the level of protein translation. The findings in this study are of interest to the field of gene therapy and may provide a creative workaround to a longstanding problem in the field. Data is provided for EIAV lentiviral vectors, which likely extends to the broader class of retroviruses. Data for demonstrating the utility and benefit other vector systems such as AAV and Ad is not provided, although using a non-toxic GFP molecule experiments indicate a significant attenuation of protein expression in producer cells and no overall effect of including TRAP-responsive sequences in these vector systems.

While the main findings of the paper are of interest and consistent with the conclusions, there are several issues we would like to see addressed.

1- The abstract provides too little information on the approach taken and the scope of the results.

Given the word limit of 150 for *Nature Communications*, we believe the abstract conveys the general problem, describes the fundamental properties of the solution (the TRiP system), demonstrates its broad utility across RNA and DNA viral vector platforms by quoting headline data. Nevertheless, we have attempted to re-work the abstract to add as much information as the word limit allows.

2- the manuscript would benefit from a more extensive, clear description of the bacterial TRAP mechanism and biological role, possibly with a schematic as a panel in Fig 1.

We have added to the introduction section to broaden the description of the natural mechanism of TRAP (also inserting an additional panel to Fig 1 as suggested), as well as additional text to expand on the scope of the work carried out.

3- The authors measure vector titer by equal volume transduction, then quantifying integration events, and show that there are dramatically fewer integration events in the CMV-toxic transgene (EIAV-COX2) context than when the TRAP/tbs are included in the transfected production plasmids (EIAV-tbsCOX2). Fig 4 indicates that there appears to be 5-7 fold more viral proteins present in the EIAV-tbsCOX2 preps than normal pEIAV-COX2 produced vectors, what is also demonstrated is that

the proteomic profiles of each prep are significantly different. It is not clear if variations in integration events are solely attributable to a reduced number of virions present, which the authors claim is the case, or if transduction efficiency is affected by the vector composition.

We have revised the text to add the clarification that the reviewer is asking for. We do not claim that COX2-mediated reduction in integration events were due *primarily* to low numbers of virions, rather that this reduction is a combination of two major factors: 1] that virions numbers were reduced by ~5-fold, and 2] the virions that *were* produced had ~50-fold less VSVG envelope compared to EIAV-tbsCOX2 vector preparations. In short, COX2 expression mediated both a minor quantitative effect on virion abundance and a major qualitative effect on virion activity (namely VSVG incorporation). We argue that these two effects of COX2 can be multiplied ($5 \times 50 = 250$ -fold), which therefore largely explains the multiple log reduction in titres. We concede that we cannot exclude the possibility that down-/up-regulation of cellular factor(s) within virions may have contributed further to reduced virion activity (indeed, this is why we performed the MS-SINQ analysis to potentially identify such a factor), and we tentatively suggest that the (80-fold) reduction in fibronectin within the EIAV-COX2 preparation could have been a contributing factor, although we have not formally pursued this hypothesis.

4- the MS-SINQ method requires a minimum of quadruplicate samples for sufficient confidence in the results, whereas the present work only uses duplicates.

We have repeated the vector virion protein profiling experiment, as required by the reviewer. This involved making 12 new EIAV vector preps and performing separate MS-SINQ analysis of each one. Whilst this enabled us to reduce the number of proteins in the three-way comparison (each common protein had to vary less than 4-fold between quadruplicates in each vector type) from ~1000 to less than 300, the results between the first experiment (using duplicates) and second experiment (using quadruplicates) were in close agreement. In essence, there were few differences in protein hits and ranking within the top ~150 proteins, which represented ~90% of the protein in the preps when comparing both experimental data sets. The repeated experiment had the effect of removing 'noise' i.e. many of the least abundant hits because these tended to be more variable across each set of quadruplicates. The additional quadruplicate analysis confirmed that the main differences between EIAV-COX2 and EIAV-tbsCOX2 vector preps was a ~5-fold reduction in Gag, a ~50-fold reduction in VSVG, and a reduction in fibronectin. This data is now presented in Table 1 for the differences in ~300 proteins. We have included a more simplified graphical representation of the average protein content of each of the three vector preps (Fig 5), to make the general point that the profile for EIAV-tbsCOX2 (repressed 'active' gene) looks more like EIAV-GFP ('inert' gene), in contrast to EIAV-COX2, which has a large 'tail', representing a large up-regulation of low-abundant proteins that are incorporated into the vector prep. We have also maintained the two physical measures of vector virions, which confirm the reduction in amount of Gag (capsid, p26) and VSVG by western blot, and F-PERT; the measure of RT activity in each prep (Supplementary fig 5b).

5- The effects of vector preps with different protein contents on target cells are difficult to interpret, and interpretation of the data would benefit from an additional physical titer titration.

We believe the point raised here is answered by data already present within the manuscript. We have added text to better clarify this. We perform two analysis of particle titration on these samples displayed in Supplementary figure 5b: 1] F-PERT (qRT-PCR), which measures the reverse transcriptase contained within virions and 2] immunoblotting of the capsid protein (p26). Both of these analyses independently confirm the exquisite sensitivity of MS-SINQ i.e. there is a reduction (~5-fold) in virion particle number when COX2 is expressed during production. The additional analysis of VSVG in virions in the same immunoblot is also consistent with the large reduction (>50-fold) in VSVG content of virions.

6- While the data provided on AAV and Ad does suggest that the relevance of this system can be extended to other vector systems, no demonstration of its benefit with a toxic transgene is provided. The manuscript would substantially increase in impact if indeed these data would demonstrate its utility across the most commonly used gene transfer vector systems. In absence of these data, we would recommend to include the present data however do not refer to those at 'proof-of-concept' in the abstract.

We have directly addressed the reviewer's request by producing both AAV and Adenovirus based vectors encoding the pro-apoptotic gene Bax. Over-expression of this protein within cells rapidly induces apoptosis, and other researchers in the AAV/Adenoviral vector field have previously reported extreme difficulty in producing vectors encoding Bax in HEK293-based cells (Kagawa, *et al*, 2000. *Gene Ther.* 7:75-79; Strobel *et al*, 2015. *Mol Ther.* 23:1582-1591); all employed some kind of control system to reduce Bax expression during vector production. We have restructured this part of the results section to focus on the production of Adeno-CMV-tbsBax-ires-GFP, and refer to production of scAAV-CMV-tbsBax within supplementary information. We found that expression of Bax within cells post-transfection with plasmid DNA or post-infection with Ad vector results in loss of GFP expression (driven by the IRES element), presumably due to induction of apoptosis. Therefore, in order to generate biological titres for Adeno-CMV-tbsBax-ires-GFP vector stocks, we had to develop a TRAP-expressing cell line in which the vectors could not replicate in order to ensure that GFP events did not represent new rounds of replication that would occur if we had used HEK293T.TRIp cells (E1-positive). Stable HeLa-TRAP cells were therefore made. Since these cells were capable of repressing Bax, the biological titre read-out was based on GFP expression, and is reported in GFP TU per mL. Therefore, the GFP TU per mL titre read-out represents a functional measure of titre. In addition direct estimation of Ad vector particle titre was made using QPCR. The data demonstrate that Adeno-CMV-tbsBax-ires-GFP could only be efficiently produced and amplified within HEK293T cells expressing TRAP. After four rounds of Ad vector passaging and end-point titration on HeLa and HeLa-TRAP cells, crude vector stocks Adeno-CMV-tbsBax-iGFP vector stocks produced in HEK293T.TRIp cells gave titres in the region of 1×10^8 GFP TU per mL, whereas no detectable vector was generated from passaging on HEK293T cells (minimum sensitivity of 1×10^4 GFP TU per). We then transduced HeLa and HeLa-TRAP cells with equivalent GFP TUs of Ad vector (Adeno-CMV-tbsGFP or Adeno-CMV-tbsBax-iGFP) and performed immunoblotting to GFP, Bax and the cleaved form of PARP, which is induced during apoptosis. This data demonstrated that the Adeno-CMV-tbsBax-iGFP vector produced not only was capable of expressing high levels of Bax but the Bax protein was functional since cleaved PARP was detectable in HeLa cells at levels substantially higher than untransduced cells. Only endogenous

levels of Bax were detected in HeLa-TRAP cells, demonstrating that the Bax expression cassette remained TRAP-repressible during vector passaging. Finally, we performed qPCR of purified vDNA from amplified vector stocks which showed that Adeno-CMV-tbsBax-iGFP vector produced in HEK293T.TriP cells achieved particle titres >150,000-fold greater than that produced in HEK293T cells, which was equivalent to Adeno-CMV-tbsGFP titress.

Reviewer #2 (Remarks to the Author):

Maunder and colleagues describe an elegant new approach to mitigate the negative impact of transgene expression in the efficiency of recombinant vector production. This bacterial translation repression system requires insertion of a small 55 nucleotide sequence (tbs) in the transcriptional cassette. The work is excellent demonstrating the effectiveness of this approach to increase lentivirus vector titers by 1-2 orders of magnitude, which is an important development for broad implementation of some of the promising ex vivo lenti-based gene therapies. This approach is also shown to be effective in repressing Cap-dependent and independent expression from bicistronic lentivirus vectors. Moreover the major changes in the proteomic content of a tbs containing lentivirus vector produced with and without TRAP are at the level of either the transgene itself or viral proteins, although fibronectin levels also appear to be higher in the presence of TRAP which may explain in part the higher transduction titers. This is an important advancement in the field of recombinant viral vector production. The data on transgene repression during adenovirus or AAV production is solid but the usefulness of this approach remains to be demonstrated in these systems which have inherently different mechanisms and limitations. It is important to consider that at least in the analyzed lentivirus/transgenes the major effect appeared to be at the level of VSV-G incorporation into the lentivirus virion.

Minor considerations:

1. It would be interesting to determine whether the overall level of VSV-G in producer cells changes or the effect of transgenes is at the level of its incorporation into virions. Same for virion proteins.

Whilst we have not carried out in depth analysis of VSVG expression in EIAV vector production cells during this work, previous reports (Radcliffe *et al*) and our unpublished work has demonstrated that VSVG is still detectable within cells when inhibited from incorporation into virions. However, the level of VSVG in cells under these conditions appears to be linked to the magnitude of the inhibitory affect, and this in turn affects steady-state levels of VSVG. So under certain circumstances VSVG levels in cells can appear to be very low. Our running hypothesis is that over-expression of a glycosylated protein inhibits the secretory pathway, leading to internalisation of VSVG away from sites of assembling virions. However, the extent of internalisation may then impact on the turn-over of VSVG within the cell. This effect is reported in Radcliffe *et al*, and we reference this in the report. We agree that COX2-induced changes within the vector production cell would be of interest but would represent a significant amount of work and cost that we believe would not add to the main focus of the manuscript.

2. At least in some preps in addition to the functional titer, the authors should also analyze the

particle content by a simple p24 ELISA as it would clarify whether the decreased titer could also be caused by inefficient packaging of the vector genome, but not a direct effect on virion assembly.

We are not entirely sure of the point being raised here. An ELISA for capsid would represent a physical measure of virions rather than genome packaging efficiency. Indeed, as mentioned in response to the other reviewer, we have already performed both F-PERT (qRT-PCR) and Western blot to EIAV capsid to measure vector particle numbers. However, biological and integration titres remain the gold standard, and most meaningful measure of titre. We have clarified the text to more clearly reflect the fact that F-PERT and Western blot analysis is directly proportional to the number of vector particles.

3. These details do not change the enthusiasm for this manuscript.

This work is considered high impact, but in this reviewer's opinion it is better suited for a specialty journal focusing on gene therapy methods. Nonetheless congratulations on an elegant system to address the production limitations that are known in the field

We believe that the additional work demonstrating proof-of-principle for the TRiPA_{Adeno} and TRiPA_{AAV} systems substantially adds to the impact of the manuscript, as argued by Reviewer 1 and the Editor. It will therefore have wide appeal to both the Gene Therapy and Vaccine fields, and consequently would be a good 'fit' in *Nature Communications*.

REVIEWERS' COMMENTS:

Reviewer #1 (Remarks to the Author):

The authors addressed the concerns raised by this reviewer in a satisfactory manner. This manuscript presents a creative and potent way to improve titers on difficult to produce viral vectors of various flavors (lenti, adeno, aav). Given the potential of this approach to unlock experimental and/or therapeutic strategies previously complicated due to transgene toxicity, this referee believes the approach is of significant impact to be published in a journal as prestigious as Nature Communications.

A remaining concern of this referee is the abundant presence of TRAP within lentiviral preparations, which may pose a safety concern in therapeutic settings that leverage this production method. The authors only minimally address this in the manuscript (in that intracameral injections do not demonstrate any evidence of antibody response). I do not believe this concern needs to be addressed experimentally in the current study, however the paragraph lines 539-545 is highly speculative. I would feel more comfortable if the authors stated the potential concern and requires further study. Furthermore, for Ad and or AAV this is likely not a problem (no envelope).

Lastly, while the authors indicate the TRAP abundance in Table 1 and in the text (line 311), the graphs in Fig5 should also specific TRAP for full transparency.

Reviewer #2 (Remarks to the Author):

The authors addressed the main concerns raised by the reviewers very effectively and provide extensive new data demonstrating unequivocally the usefulness of this elegant TRiP system to enhance production of a broad range of gene therapy vectors. This is an important technical development as vector based gene therapy moves rapidly into human application and production scales become more pressing.

Response to reviewer's final comments: manuscript NCOMMS-16-02222(A)

(Reviewers' comments are stated in order and authors' responses are in bold)

Reviewer #1 (Remarks to the Author):

The authors addressed the concerns raised by this reviewer in a satisfactory manner. This manuscript presents a creative and potent way to improve titers on difficult to produce viral vectors of various flavors (lenti, adeno, aav). Given the potential of this approach to unlock experimental and/or therapeutic strategies previously complicated due to transgene toxicity, this referee believes the approach is of significant impact to be published in a journal as prestigious as Nature Communications.

A remaining concern of this referee is the abundant presence of TRAP within lentiviral preparations, which may pose a safety concern in therapeutic settings that leverage this production method. The authors only minimally address this in the manuscript (in that intracameral injections do not demonstrate any evidence of antibody response). I do not believe this concern needs to be addressed experimentally in the current study, however the paragraph lines 539-545 is highly speculative. I would feel more comfortable if the authors stated the potential concern and requires further study. Furthermore, for Ad and or AAV this is likely not a problem (no envelope). Lastly, while the authors indicate the TRAP abundance in Table 1 and in the text (line 311), the graphs in Fig5 should also specific TRAP for full transparency.

We have amended the manuscript and Fig 5 to reflect the concerns raised by the reviewer. We have altered the discussion section:

“The lack of antibody response to TRAP in rats treated intracamerally with EIAV-tbsCOX2 in the eye (an organ/route known to be particularly sensitive to inflammation) provides evidence that the presence of TRAP protein within retroviral vector virions is unlikely to be problematic. In support of this, it is our experience in the clinic that human antibody responses to protein in purified VSV-G pseudotyped lentiviral vector preparations were only found at the highest doses, and were primarily to the viral glycoprotein, which is present on the virion surface⁴⁴. Nevertheless, future investigation into potential pre-existing anti-TRAP responses in human sera may be warranted for application of the TRiPRetro/Lenti platform.”

It is correct that, without a formal study, we can only speculate as to the true pre-existing immunity to TRAP within the human population. This is also true for any other gene control system utilising a non-human repressor protein such as tetR or lacR. However, we reference our previous work (ref #44), where the main antibody response to vector product in humans was to VSV-G (and only at the highest dose), in order to suggest that surface display of non-human proteins on virions is more likely to induce an immune response compared to those found inside virions. Given that TRAP is not a membrane protein, we don't believe this concept to be highly speculative but a logical one (albeit untested).

Reviewer #2 (Remarks to the Author):

The authors addressed the main concerns raised by the reviewers very effectively and provide extensive new data demonstrating unequivocally the usefulness of this elegant TRiP system to enhance production of a broad range of gene therapy vectors. This is an important technical development as vector based gene therapy moves rapidly into human application and production scales become more pressing.